# Formulation of Rosemary Extracts through Spray-Drying Encapsulation or Emulsification

Lamprini-Danai Kanakidi [1], Dimitrios Tsimogiannis [1], Sotirios Kiokias [2] and Vassiliki Oreopoulou [1,*]

[1] Laboratory of Food Chemistry and Technology, School of Chemical Engineering, National Technical University of Athens, 15780 Athens, Greece; danae.kanakidi@gmail.com (L.-D.K.); ditsimog@chemeng.ntua.gr (D.T.)

[2] European Research Executive Agency, Place Charles Rogier 16, 1210 Bruxelles, Belgium; sotirios.kiokias@ec.europa.eu

\* Correspondence: vasor@chemeng.ntua.gr

**Abstract:** Natural, plant-derived antioxidants can be used to prolong the shelf life of food or cosmetics, or as health-promoting additives. Although their extraction from plants has been extensively studied, purification and formulation processes need further research to allow their exploitation. In the present work, rosemary extracts were obtained by successive extractions with acetone and water or single extraction by either an acetone:water or ethanol:water mixture. The extracts were analyzed by HPLC-DAD, and rosmarinic acid, carnosic acid, carnosol, and several flavonoids were identified and quantified. The extracts obtained by water or aqueous mixtures of organic solvents were encapsulated in maltodextrin combined with gum arabic with a high encapsulation yield (90–100%) and efficiency (97%) for rosmarinic acid and flavonoids. The acetone extract, rich in carnosic acid, was transformed to oil solution and either encapsulated or formulated in emulsion. The shelf life of encapsulated products was tested over a period of six months, and the results showed high retention of rosmarinic acid (88%) and lower of flavonoids (54–80%). Carnosic acid presented lower retention either encapsulated in solid powder (65–70% after one month at ambient temperature) or in emulsion (48% after 20 days of storage at 15 °C), while it was partially transformed to carnosol.

**Keywords:** rosemary; extraction; spray drying; emulsion; storage stability; carnosic acid; rosmarinic acid; carnosol; flavonoids

## 1. Introduction

Natural phenolic antioxidants extracted from various plant sources are constantly gaining growing interest of researchers and food or cosmetic companies [1–4]. Among them, antioxidants from rosemary are broadly recognized and have received the EU approval for use in food (E-392, European Union directives 2010/67/EU and 2010/69/EU), and their market is expected to raise up to 260 million US$ by 2024 [5]. Rosemary is one of the few plants (including common *Salvia* species) that contains carnosic acid and carnosol as their main antioxidant components [6]. In addition, the plant is rich in rosmarinic acid, which is a potent radical scavenger and antioxidant [7,8]. Rosmarinic acid is present in some herbs of the *Lamiaceae* family and is responsible for the high antiradical and antioxidant activities of the extracts obtained from these herbs [6]. Other phenolic components of the plant comprise flavonoids, mainly in the form of glycosides [6].

Most research efforts examined the extraction of antioxidants by using organic solvents, such as acetone, ethanol, and methanol, or their mixtures with water [6–8]. Researchers focused mainly on the quantitative recovery of rosmarinic acid, carnosic acid, and carnosol, as they are the main antioxidant components of rosemary [7–10]. Conventional, solid/liquid extraction is a broadly used method, while novel procedures, in particular, ultrasound-assisted extraction (UAE), microwave assisted extraction (MAE), and accelerated solvent extraction (ASE), have been examined [6,8–13].

To further optimization of the extraction conditions, purification and formulation in appropriate carriers is necessary for the commercial exploitation of antioxidants. Formulation should protect the antioxidant compounds during storage, facilitate their handling and addition to foods or cosmetics, and possibly control their release in the final product [14,15]. Non-toxic carriers, suitable for foods or cosmetics, and imparting no taste or odor to the final product, must be used [15]. Dried products have several advantages over liquid ones, such as easier handling, transport, and storage. In addition, they can be used in the manufacture of solid dosage forms (e.g., tablets and capsules) that represent most of the food supplements or medicines manufactured worldwide [14,16]. However the published works about phenolic antioxidants formulation are limited [16–19]. Couto et al. [16] and Chaul et al. [17] proposed the drying of rosemary antioxidants, through spray drying without any additive, and examined the process conditions, such as inlet air temperature, extract feed rate, and airflow rate. Although the obtained power exhibited antiradical activity, an appreciable amount (around 50%) of total polyphenols was lost [16]. To protect the thermo-sensitive bioactive compounds during processing and also storage, several researchers examined the encapsulation in appropriate coating materials. Several coating materials may be used, but maltodextrin is the most commonly used, due to its ability to form film, water solubility accompanied with low viscosity even at high concentration, and low cost [19]. The encapsulation efficiency of maltodextrin might be improved by the addition of other polysaccharides, such as gum arabic or pectin [18,19]. Spray drying is the most commonly used encapsulation technique because it is technologically developed, has low cost, and is easily applicable at industrial scale [14,15,20]. Spray drying equipment for several food applications is available, able to process a wide range of materials, in continuous mode, and produce stable, good quality dried products [14,15]. In spray drying encapsulation, almost entirely, aqueous feed dispersions are used. Therefore, the water solubility of the coating material is of major importance, thus rendering maltodextrin one of the most suitable wall materials [15].

Another potent formulation of lipophilic antioxidants is emulsification. In many cases, it is advantageous to deliver lipophilic antioxidants in aqueous medium because this enables their incorporation in some foods (beverages, liquid products) and may increase their desirability, palatability, and bioactivity [21]. Moreover, these ingredients can be easily absorbed by the skin in emulsified forms, especially as nanoemulsions [22].

In a previous paper of this research team, mixtures of acetone or ethanol with water, and, in particular, acetone:water 80:20, *v/v*, and ethanol:water 60:40, *v/v*, were shown to be the most effective for the quantitative extraction of rosemary antioxidants, while UAE enhanced the antioxidants recovery, compared to conventional solid/liquid extraction [8]. In addition to following UAE with the aforementioned solvents, in the present work, we used a fixed bed extraction procedure and performed either successive extractions with water and acetone, in order to separate polar (phenolic acids, flavonoids) from non-polar (carnosic acid, carnosol) compounds, or a single extraction with acetone:water 80:20, *v/v*, to be compared to the UAE. Non-polar antioxidants are completely soluble in lipid substrates, while more polar ones can be used in different formulations (i.e., aqueous phase of emulsions). Fixed bed extraction offers the advantage of no need to separate liquid from the solid phase, while performing successive extractions, until the exhaustion of the solutes from the solid matrix. The procedure has not been examined for the extraction of antioxidants from rosemary and could provide an easily applicable alternative for industrial application. The extracts were formulated to powders, through encapsulation by spray drying after the removal of the solvents. The spray drying encapsulation of rosemary antioxidants that are thermo-sensitive (especially carnosic acid) has not been reported in literature. Moreover, the acetone extract (rich in carnosic acid and carnosol) was incorporated in an oil-in-water (o/w) emulsion. Such a model system can be the basis for the formulation of novel cosmetic creams or liquid and emulsified food products and offer new innovative pathways for the associated industrial sectors. The products were further studied during storage to determine their stability. The aim of the present work

was (i) to apply effective extraction techniques in order to obtain rosemary extracts rich in either phenolic acids and flavonoids or in carnosic acid and carnosol, as well as in all the aforementioned antioxidants; (ii) to formulate the obtained extracts in encapsulated powders or emulsions that could be used in various food or cosmetic formulations; and (iii) to examine the storage stability of the products.

## 2. Materials and Methods

### 2.1. Solvents and Reagents

Ethyl alcohol and acetone (analytical reagents, Sigma-Aldrich, Steinheim, Germany) were used for the extraction. Folin Ciocalteu phenol reagent (2N, Merck, Darmstadt, Germany), sodium carbonate anhydrous (>99%, Mallinckrodt, St. Louis, MO, USA), gallic acid (98%, Acros Organics, Fair Lawn, NJ, USA) 2,2-diphenyl-1-picryl hydrazyl (DPPH radical, Sigma-Aldrich, Steinheim, Germany), and (S)-(-)-6-hydroxy-2,5,7,8-tetramethyl chroman-2-carboxylic acid (Trolox, Aldrich, Milwaukee, WI, USA) were used for the analysis of the extracts. Water, methanol, isopropanol, 1-butanol, acetonitrile, orthophosphoric, and trifluoroacetic acid, obtained from Fisher Chemical (Leicestershire, UK), were used for HPLC-DAD analyses, and the standard compounds were carnosic acid (Dayang Chemicals Co., Hangzhou, China), carnosol (Extrasynthese, Lyon, France), rosmarinic acid (Sigma-Aldrich, Steimheim, Germany), caffeic acid (Sigma-Aldrich, Steimheim, Germany), and quercetin dihydrate (Sigma-Aldrich, Steimheim, Germany). The materials used for the encapsulations were maltodextrin (MD) 18-20 DE, obtained from Astron Chemicals SA (Attica, Greece), and gum Arabic (GA), from Nexira Food (Rouen, France). Tween 20 (Fisher Scientific, Waltham, MA, USA), and distilled monoglycerides Type P(V) Fine Powder (Rikevita, Malaysia) were used as emulsifiers. A medium-chain triglyceride oil (MCT), consisting of capric/caprilic acids, was obtained from Cosmochem SA (Attica, Greece). As preservative, a dilute solution of sodium azide (0.5‰) obtained from Acros Organics (Morris Plains, NJ, USA) was used.

### 2.2. Plant Material

The raw material used in the experiments was rosemary from experimental cultivations of the Hellenic Agricultural Organization "DEMETER". The harvest took place in May 2019. The plant was subjected to water-steam distillation for 6 h, so as to remove the essential oil, as described by Tsimogiannis et al. [23]. The wet herbal residue was dried at 35 °C for 24 h in a ventilated oven (Function Line UT20, Heraeus Instruments GmbH, Hanau, Germany), and a high-speed blender was used in order to grind the dry plant.

### 2.3. Extraction Procedures

#### 2.3.1. Fixed Bed Extraction (FBE)

The fixed bed extraction was carried out with an experimental device consisting of a stainless steel, cylindrical extractor with an inner diameter of 3 cm and a height of 9 cm. The material (20.0 g) was placed in the cylindrical extractor, and plastic tubes were applied to the inlet and outlet of the extractor, through which the solvent passed to wet the plant material. A peristaltic pump (Millipore, Bedford, MA, USA) was used to regulate the solvent flow (2.0 mL/min). The solvent entered from the inlet of the extractor located at the bottom, passed through the plant material, and exited from the top of the extractor, where it was collected in a volumetric cylinder. It was adjusted to enter from the bottom, up to achieve uniform wetting of the plant material. Pure acetone, deionized water, and a mixture of 80:20, acetone:water, *v/v*, were used as solvents in the experiments, with the addition of orthophosphoric acid at a content of 0.1% by volume of the organic solvent, in order to protect the carnosic acid [8]. The extraction ended with the collection of 200 mL of extract, as preliminary experiments showed that, at this point, the maximum extraction efficiency was achieved.

### 2.3.2. Ultrasound Assisted Extraction (UAE)

The UAE was performed into a spherical extraction vessel equipped with a multiple-neck lid bearing a vertical water cooler. The material (5.0 g) was placed in the vessel, and 100 mL of solvent was added, so that the final solid-to-liquid ratio was 1:20 g/mL, which proved efficient for the recovery of phenolic antioxidants [24]. The vessel was immersed in an ultrasonic bath (Elmasonic S, Elma, Schmidbauer, Germany), and the ultrasonic frequency was adjusted to 37 kHz. The temperature of the bath was maintained at 22 °C, and the extraction was carried out for 1h, as it had previously been shown that equilibrium of phenolic compounds extraction was reached at 1 h [8]. The extractions were carried out with a mixture of acetone:water (80:20) or ethanol:water (60:40). In all solvents, orthophosphoric acid (0.1%, $v/v$) was added to retain the carnosic acid at the final extracts.

### 2.4. Spray Drying Encapsulation

The water extract was partially concentrated in a rotary evaporator (Heidolph G1 Hei-VAP Value, Schwabach, Germany), under vacuum, and adjusted to a volume of 50 mL, before the encapsulation by spray drying. The rotary evaporator was also used for the removal of acetone or ethanol from the acetone-water and the ethanol-water extract, respectively, the evaporation was continued for the removal of part of the water, and the concentrate was adjusted to 50 mL final volume. In order to encapsulate the acetone extract, it was first transformed into an oil solution, with the simultaneous removal of the solvent. For this purpose, the acetone extract was added to the required amount of capric/caprylic acid triglyceride (MCT), so as to obtain a final concentration of 2.50 g carnosic acid plus carnosol/kg oil solution, because rosemary extracts are commonly expressed and commercialized according to their carnosic acid plus carnosol content [25]. The solvent was subsequently removed from the oil-extract mixture, in the rotary evaporator (Heidolph G1, Hei-VAP Value, Schwabach, Germany), under vacuum, at room temperature.

The extracts or the oil solution consisted the core material in the feed solution, and their content was 5 or 10% (g dry extract or oil solution/100 g solids in the solution). The wall material consisted of MD or a combination of MD with GA. The total solid concentration in the feed solution was maintained constant at 30% (g solids/100 g solution), whereas the feed solution mass equal to 125 g, during all the experiments.

For the preparation of the wall-material solution, the appropriate amount of MD was added to distilled hot (50 °C) water under stirring that was continued until the solution became clear. In case of MD-GA combinations, initially, the appropriate amount of GA was added to distilled hot water. The temperature was maintained below 50 °C and the GA solubilized under continuous stirring. When complete GA dissolution was obtained, the appropriate amount of MD was added, and stirring was continued until the solution became clear. In case of water extract or organic mixture extracts encapsulation, the extract, after solvent removal and concentration as mentioned above, was added, at the appropriate content, and the feed solution was incubated overnight at 4 °C, to achieve better hydration of the encapsulation carriers. The hydrated mixture, after reaching room temperature, was homogenized by using a high-speed homogenizer (UnidriveX 1000, CAT, Ballrechten-Dottingen, Germany) for 15 min, at 15,000 rpm.

In case of the encapsulation of the oil solution containing the acetone extract, the wall material was incubated overnight, as mentioned above, and Tween 20 in a content of 0.1% of the total solids was added as emulsifier before homogenization. The mixture was transferred to the high-speed homogenizer, and the oil solution was added slowly during the homogenization.

Each homogenized mixture was fed to the spray dryer (Büchi B-191 Mini, Büchi Labortechnik AG, Flawil, Switzerland) under continuous stirring in order to maintain its homogeneity. The atomization and spray-drying process parameters were:

i.  air inlet temperature: 140 or 160 °C, selected according to literature [18,19],
ii. air outlet temperature: 70–85 °C when the inlet temperature was 140 °C, and 85–100 °C when the inlet temperature was 160 °C,

iii.    atomization pressure: 5 bar, based on preliminary experiments, and
iv.    feed solution temperature: 25 °C and the feed flow rate was adjusted and remained constant at the point where the desired output temperature was reached, according to preliminary experiments.

The product, a yellowish powder, was collected and part of it was used for analysis, while the rest was stored in a multilayer package of OPP 20 μm/Adhesive/PET MET 12 μm/Adhesive/PE 75 μm in dimensions 12.5 × 16.5 cm (Vlachos Bros SA, Athens, Greece). The package was sealed airtight and kept at room temperature, until opened for analysis.

*2.5. Emulsions Preparation and Storage Test*

The oil solution obtained from the acetone extract (as described in Section 2.4) was used to prepare an oil-in-water emulsion with 80 g/kg lipid phase, 15 g/kg hydrophilic emulsifier (Tween 20), 15 g/kg lipophilic emulsifier ("Distilled monoglycerides"), and 0.1 g/kg antimicrobial agent. More specifically, the emulsifiers (6 g Tween 20 and 6 g Distilled monoglycerides) were dissolved in hot water (276 g) under continuous stirring. After cooling the mixture, 80 g of dilute 0.5 g/kg sodium azide antimicrobial solution was added. The mixture was transferred to a high-speed homogenizer (UnidriveX 1000, CAT, Ballrechten-Dottingen, Germany), operating at 15,000 rpm, and 32 g of the previously prepared oil solution was added dropwise and under stirring. Homogenization lasted 15 min. The pre-emulsion was then treated by a high-pressure homogenizer (Invensys APV-1000, London, UK), at 700 bar and 10 passes, in order to achieve good emulsification according to preliminary experiments.

The emulsion was separated to batches (10 g) that were placed in open vials in order to study the shelf life of its components. The vials were stored at either 15 °C or 37 °C (accelerated aging experiment). Duplicate vials were removed at definite time intervals and processed for analysis of the components.

*2.6. Analytical Methods*

The collected extracts were kept under refrigeration (4 °C) and analyzed within 24 h. The total phenol content, the antiradical activity, and the solid residue were determined, and the selectivity of the extraction was calculated as described below. All measurements were repeated in duplicate and averaged.

2.6.1. Determination of Total Phenol Content (TPC)

The total phenolic content (TPC) was determined by the Folin–Ciocalteu method, as described by Singleton et al. [26]. The absorbance was measured at 765 nm using a UV-Vis spectrophotometer (Hitachi U2900, Hitachi High-Technologies Corporation, Tokyo, Japan). A calibration curve with gallic acid reference compound was constructed, and the results are expressed as gallic acid equivalents on the dry plant basis (g GAE/kg dw).

2.6.2. DPPH Free Radical Scavenging Assay

The antiradical activity was determined by the DPPH radical assay according to the methodology reported by Brand-Williams et al. [27]. Samples (0.1 mL) of each extract were added to 3.9 mL of freshly prepared DPPH radical solution in methanol ($6 \times 10^{-7}$ M), and the absorbance at 515 nm was recorded after 30 min, by using a UV-Vis instrument (Hitachi U2900). The results are expressed as Trolox equivalents on dry plant basis (g Trolox/kg dw), through the construction of a calibration curve obtained with Trolox as a standard compound.

2.6.3. Solid Residue and Selectivity

The solid residue was determined, by drying duplicate samples (5 mL) of each extract in an oven at 103 °C and expressed as g/$L_{extract}$.

The selectivity (%) of the extraction is defined as the percentage of the total phenol content (TPC, expressed on extract volume basis) on the solid residue of the extract. The unit of measurement is [g GAE/g total solids] $\times$ 100 (%)

### 2.6.4. Encapsulation Yield and Efficiency

For the determination of the encapsulation yield (EY) and encapsulation efficiency (EE), the method of Mahdavi et al. [28] was followed, with some modifications. To determine the total content (TC) of the antioxidant compounds in the encapsulated powder produced by the spray drier, 1 g of powder was mixed with 10 mL of deionized water. The sample was stirred with a vortex apparatus for about 30 s to dissolve thoroughly. Ethanol (15 mL) was then added and stirred for 5 min, to precipitate the wall material. The mixture was filtered through a micro filter (0.45 μm). The obtained liquid was analyzed by HPLC-DAD. The EY (%) was determined as the ratio of each compound in the microcapsule (dry basis) to its content in the feed solids (calculated according to the consistency of the extract or the oil solution, and the core content used in each treatment).

To determine the surface content (SC), i.e., the content of each compound on the surface of the encapsulated powder, the method of Tolun et al. [18] was used, with some modifications. One gram of powder was washed with 10 mL of a mixture of ethanol:methanol (1:1, $v/v$) and was stirred using a vortex for 1 min. The sample was then filtered through a micro filter (0.45 μm), and the liquid was analyzed by HPLC-DAD. In this way, the content inside the microcapsule can be determined by subtraction, i.e., TC-SC, and, consequently, the EE (%), by using the following equation: [(TC-SC)/TC] $\times$ 100%.

### 2.6.5. Determination of Powder Moisture Content

In order to determine the moisture of the powders, duplicate samples (0.5 g) of each one were dried in an oven at 103 °C until constant weight. The moisture content was expressed on wet basis as moisture content (%) = $(m_{initial} - m_{final})/m_{initial} \times 100$.

### 2.6.6. Emulsion Shelf Life Analysis

Duplicate samples of the emulsions removed at definite time intervals were analyzed by HPLC-DAD to determine their consistency in antioxidant compounds. For the analysis, 2 mL of emulsion were diluted in a 10 mL flask with isopropanol solution containing 0.1% $H_3PO_4$. As storage time increased, and, consequently, degradation of emulsion components proceeded, the emulsion sample was increased to 3 mL. The solution was immediately used for the analysis.

### 2.6.7. HPLC-DAD

For the determination of the main phenolic compounds of each extract the high-performance liquid chromatography with a diode-array detector (HPLC-DAD) method, described by Psarrou et al. [8], was used. The instrument consisted of an HP 1100 gradient pump and a diode array detector (Hewlett Packard, Waldbronn, Germany), which was connected to a ZORBAX Eclipse XDB-C18 column (5 μm, 250 $\times$ 4.6 mm, Agilent, Santa Clara, CA, USA), thermostatted at 30 °C. The solvent system comprised water (A), methanol (B), and acetonitrile (C), each containing 0.2% trifluoracetic acid. The initial composition of the mobile phase was 90% A, 6% B, and 4% C and changed with linear gradients to 85% A, 9% B, and 6% C within 5 min, 71% A, 17.4% B, and 11.6% C within 30 min, and 0% A, 85% B, and 15% C within 60 min. The solvent flow rate was 1 mL/min, and the injection volume was 20 μL. Detection was monitored at 280 and 360 nm, and the data were processed by the Varian Workstation (Varian Inc., Palo Alto, CA, USA). In particular, for the quantification of the phenolic diterpenes, isocratic elution with a mobile phase consisting of water-acetonitrile 40:60 acidified with 0.1% phosphoric acid was used, and the detection was monitored at 230 nm. The quantification of analytes in the samples was based on the respective calibration curves obtained with rosmarinic acid, caffeic acid, and quercetin, using the above gradient method, as well as carnosic acid and carnosol, using the isocratic method.

*2.7. Statistical Analysis*

Experiments were carried out at least in duplicate, and, during analysis, each measurement was repeated twice. Results were averaged (mean values ± standard deviation) and statistically analyzed with one-way ANOVA ($p < 0.05$) by use of the software Statistica 7.0 (Statsoft, Tulsa, OK, USA). Statistical differences among the various treatments were calculated by post-hoc comparison of means according to Duncan's multiple range test.

## 3. Results

*3.1. Extraction Yield and Phenolic Compounds Content in the Extracts Obtained by Different Procedures*

Table 1 presents the TPC, antiradical activity, and selectivity of the extracts obtained by different extraction methods or solvents, while Table 2 presents the compounds identified and quantified by HPLC-DAD in each extract. When performing successive extractions with acetone followed by water, acetone extracted considerably lower amounts of total phenols than water, while the acetone-water mixture yielded approximately the sum of the TPC obtained by successive extractions. The application of UAE did not result in significantly higher amounts of TPC yield with either solvent mixture (Table 1). With regard to the activity against the DPPH radical, FBE exhibited the highest values, while the values obtained by successive extractions followed more or less the TPC yields. The extracts obtained by UAE showed lower antiradical activity, especially the one obtained with the alcohol-water mixture, possibly due to degradation of some active compounds by sonication. The selectivity values (i.e., the percentage of TPC versus the total solids content of the extracts) obtained from all extracts made with the solvent mixtures were similar regardless of the type of mixture. Compared to the selectivity values obtained with water or acetone, these from solvent mixtures were higher.

**Table 1.** The effect of different extraction methods or solvents on the obtained total phenol content (TPC), antiradical activity and selectivity of the extraction.

| Extraction | TPC Yield (g GAE/kg dw) [1] | Antiradical Activity (g Trolox/kg dw) [1] | Selectivity (%) |
|---|---|---|---|
| Acetone 100% (FBE [2]) | 17.5 ± 1.4 [a] | 27.2 ± 2.9 [a] | 14.8 ± 1.2 [a] |
| Water 100% (FBE) | 44.5 ± 3.8 [b] | 76.3 ± 6.1 [bc] | 19.6 ± 1.7 [b] |
| Acetone:water 80:20 (FBE) | 60.0 ± 2.6 [c] | 127.3 ± 4.5 [d] | 24.4 ± 0.6 [c] |
| Acetone:water 80:20 (UAE [3]) | 64.5 ± 6.0 [c] | 91.6 ± 7.7 [c] | 26.9 ± 2.5 [c] |
| Ethanol:water 60:40 (UAE) | 62.5 ± 4.2 [c] | 63.2 ± 5.5 [b] | 26.3 ± 1.8 [c] |

[1] The results are expressed on the dry plant basis, [2] fixed bed extraction, [3] ultrasound assisted extraction, superscript letters in the same column indicate significant differences ($p < 0.05$).

The HPLC analysis revealed that, when successive extractions were applied, the phenolic diterpenes, which are non-polar compounds, were exclusively extracted with acetone (Table 2). The most abundant phenolic diterpene was carnosic acid, followed by carnosol, while three other peaks were separated. The major among them was identified as methyl carnosate, according to its retention time and UV-Vis spectrum. The highest total phenolic diterpenes yield was obtained by acetone:water 80:20. The use of ethanol:water 60:40 in UAE resulted in partial oxidation of carnosic acid to carnosol and decrease of the overall phenolic diterpenes yield.

Pure acetone extracted also small amounts of rosmarinic acid and flavonoids, while their quantitative recovery was obtained by the following extraction with water (Table 2). Among the flavonoids, acetone extracted higher quantities of the aglycones (i.e., hispidoulin, ladanein, genkwanin, salvigenin, 4-methoxytectochrysin), while water recovered higher amounts of the glycosides (nepitrin, homoplantaginin, isoscutellarein). In addition to the compounds presented in Table 2, the HPLC analysis showed the presence of four more peaks belonging to the flavone or flavonol subgroups of flavonoids, according to their spectra, but could not be identified. These compounds were quantified as quercetin

equivalents and have been added into the total flavonoids sum presented in Table 2. In addition, traces of salvianolic acid were detected (not quantified), mainly in the water extract. The acetone-water mixture recovered approximately the sum of the compounds extracted by the successive extractions, while UAE with either solvent mixture did not enhance the extraction of individual phenolic compounds, compared to FBE.

**Table 2.** Quantification of identified compounds by HPLC-DAD in each extract.

| Identified Flavonoids | Acetone Extract (FBE [1]) C (g/kg dw [3]) | Water Extract (FBE) C (g/kg dw) | Acetone:Water 80:20 (FBE) C (g/kg dw) | Acetone:Water 80: (UAE [2]) C (g/kg dw) | Ethanol:Water 60:40 (UAE) C (g/kg dw) |
|---|---|---|---|---|---|
| **Phenolic diterpenes** | | | | | |
| carnosic acid | 9.5 ± 0.5 | – | 13.9 ± 0.5 | 12.7 ± 0.5 | 7.0 ± 0.3 |
| carnosol | 2.1 ± 0.2 | – | 2.4 ± 0.1 | 2.8 ± 0.1 | 5.0 ± 0.3 |
| other phenolic diterpenes | 1.5 ± 0.0 | – | 1.9 ± 0.2 | 1.9 ± 0.1 | 1.7 ± 0.1 |
| Total phenolic diterpenes | 13.1 ± 0.7 | – | 18.2 ± 0.3 | 17.4 ± 0.7 | 13.7 ± 0.7 |
| **Phenolic acids** | | | | | |
| caffeic acid | – | 1.2 ± 0.1 | 1.3 ± 0.1 | 1.14 ± 0.1 | 1.04 ± 0.1 |
| rosmarinic acid | 3.2 ± 0.3 | 16.3 ± 1.7 | 21.3 ± 1.2 | 20.7 ± 1.4 | 18.0 ± 0.2 |
| **Flavonoids** | | | | | |
| Nepitrin [4] | 1.6 ± 0.1 | 9.5 ± 0.9 | 13.4 ± 1.4 | 13.7 ± 1.8 | 12.2 ± 0.2 |
| Homoplantaginin [4] | 0.8 ± 0.1 | 5.0 ± 0.6 | 4.5 ± 0.1 | 5.1 ± 0.1 | 4.9 ± 0.4 |
| Isoscutellarein [4] | 1.0 ± 0.1 | 6.1 ± 1.0 | 5.8 ± 0.7 | 8.4 ± 1.0 | 7.3 ± 0.9 |
| Hispidoulin [4] | 0.4 ± 0.0 | 0.2 ± 0.0 | 1.1 ± 0.0 | 0.7 ± 0.0 | 0.6 ± 0.0 |
| Ladanein [4] | 0.7 ± 0.0 | 0.1 ± 0.0 | 1.4 ± 0.1 | 1.1 ± 0.0 | 1.1 ± 0.0 |
| Genkwanin [4] | 0.2 ± 0.0 | 0.1 ± 0.0 | 0.6 ± 0.0 | 0.2 ± 0.0 | 0.2 ± 0.0 |
| Salvigenin [4] | 2.3 ± 0.2 | 0.1 ± 0.0 | 3.4 ± 0.1 | 2.8 ± 0.0 | 2.6 ± 0.1 |
| 4′-methoxytectochrysin [4] | 0.6 ± 0.0 | – | 0.8 ± 0.0 | 0.6 ± 0.0 | 0.4 ± 0.0 |
| Total flavonoids [4] | 9.4 ± 0.6 | 27.4 ± 2.6 | 35.6 ± 3.1 | 37.4 ± 1.6 | 33.3 ± 1.8 |

[1] Fixed bed extraction, [2] ultrasound assisted extraction, [3] The results are expressed on the dry plant basis, [4] quantified as quercetin equivalents.

### 3.2. Encapsulation of the Extracts and Storage Stability of the Encapsulated Products

3.2.1. Encapsulation Yield and Efficiency

The results of the encapsulation through spray drying, expressed as encapsulation yield and encapsulation efficiency, are presented in Tables 3–5 for the water, mixtures of water with organic solvents, and acetone extract, respectively.

**Table 3.** Encapsulation yield (EY) and encapsulation efficiency (EE) of phenolic compounds from water extracts with maltodextrin (MD) and gum arabic (GA) as wall material in different proportions (*w/w*), different inlet air temperature, constant core content 10%, and total solids content in the feed mixture 30%.

| Wall (MD:GA) | Inlet Air Temperature (°C) | Powder Moisture (%) | EY (%) | | EE (%) | |
|---|---|---|---|---|---|---|
| | | | Rosmarinic Acid | Total Flavonoids | Rosmarinic Acid | Total Flavonoids |
| 1:0 | 140 | 3.3 ± 0.0 [a] | 95.4 ± 4.5 [a] | 84.2 ± 4.5 [a] | 96.9 ± 0.2 [a] | 96.5 ± 0.2 [a] |
| 4:1 | 140 | 4.2 ± 0.3 [b] | 100.0 ± 3.3 [a] | 95.3 ± 5.4 [ab] | 96.8 ± 1.5 [a] | 97.2 ± 0.5 [ab] |
| 4:1 | 160 | 3.1 ± 0.0 [a] | 100.0 ± 4.4 [a] | 96.9 ± 0.0 [b] | 97.9 ± 0.1 [a] | 97.4 ± 0.2 [b] |
| 2:1 | 140 | 4.1 ± 0.1 [b] | 98.9 ± 0.1 [a] | 90.9 ± 3.2 [ab] | 97.4 ± 0.3 [a] | 97.6 ± 0.2 [b] |

Superscript letters in the same column indicate significant differences ($p < 0.05$).

The results presented in Table 3 show that the addition of GA to MD had no significant effect on the emulsification yield of either rosmarinic acid or total flavonoids, although the ratio MD:GA of 4:1 presented the highest values. A further increase of the GA content (i.e., MD:GA = 2:1) had no effect. The increase of inlet air temperature from 140 to 160 °C, tested at the best MD:GA value (4:1), did not differentiate the results, indicating that, within this range, no degradation of the thermo-sensitive phenolic compounds occurs. The emulsification efficiency was higher than 96% in all trials, with no significant difference among them. The moisture content of the obtained powder increased by the addition of GA. On the contrary, the increase of inlet air temperature and, consequently, of outlet temperature decreased the moisture content. These results suggest that the water extract

of rosemary, rich in phenolic acids and flavonoids, can be effectively encapsulated in an MD:GA, 4:1, wall material, with a core content of 10%, total feed solids of 30%, and inlet air temperature equal to 140 °C.

**Table 4.** Encapsulation yield (EY) and encapsulation efficiency (EE) of phenolic compounds from the extracts obtained with mixtures of organic solvents and water, with MD and GA as wall material in different proportions, constant inlet air temperature 140 °C, constant core content 10%, and total solids content in the feed mixture 30%.

| Extract | Wall (MD:GA) | Core (%) | Powder Moisture (%) | Rosmarinic Acid | Total Flavonoids | Carnosic Acid | Carnosol | Other Phenolic Diterpenes |
|---|---|---|---|---|---|---|---|---|
| | | | | | **EY (%)** | | | |
| Acetone:Water 80:20 | 1:0 | 10 | $3.8 \pm 0.5$ [a] | $98.7 \pm 0.4$ [a] | $86.3 \pm 0.1$ [a] | $57.8 \pm 1.1$ [a] | $93.2 \pm 6.8$ [a] | $67.6 \pm 4.8$ [a] |
| Acetone:Water 80:20 | 4:1 | 10 | $3.9 \pm 0.0$ [a] | $100.0 \pm 0.2$ [a] | $91.3 \pm 0.0$ [b] | $62.3 \pm 0.6$ [b] | $98.0 \pm 1.5$ [ab] | $67.2 \pm 0.8$ [a] |
| Ethanol:Water 60:40 | 4:1 | 10 | $4.2 \pm 0.1$ [a] | $99.1 \pm 1.2$ [a] | $95.5 \pm 5.0$ [b] | $65.7 \pm 1.2$ [c] | $100.0 \pm 0.8$ [b] | $94.9 \pm 3.8$ [b] |
| | | | | | **EE (%)** | | | |
| Acetone:Water 80:20 | 1:0 | 10 | $3.8 \pm 0.5$ [a] | $92.4 \pm 0.0$ [a] | $90.4 \pm 0.0$ [a] | $26.4 \pm 2.8$ [a] | $64.5 \pm 3.9$ [a] | $35.2 \pm 4.6$ [a] |
| Acetone: Water 80:20 | 4:1 | 10 | $3.9 \pm 0.0$ [a] | $95.1 \pm 0.5$ [b] | $91.8 \pm 0.2$ [b] | $31.8 \pm 0.1$ [b] | $43.5 \pm 2.1$ [b] | $37.7 \pm 0.8$ [a] |
| Ethanol: Water 60:40 | 4:1 | 10 | $4.2 \pm 0.1$ [a] | $97.2 \pm 0.7$ [c] | $91.5 \pm 0.9$ [b] | $34.3 \pm 0.7$ [b] | $66.7 \pm 1.1$ [a] | $53.0 \pm 1.9$ [b] |

Superscript letters in the same column indicate significant differences among EY or EE values ($p < 0.05$).

**Table 5.** Encapsulation yield (EY) and encapsulation efficiency (EE) of phenolic diterpenes present in the acetone extract. Spray drying parameters: wall material MD:GA 2:1, inlet air temperature 140 °C, total solids content in the feed 30%. Superscript letters in the same column indicate significant differences ($p < 0.05$).

| Core (%) | Powder Moisture (%) | EY% | | | EE% | | |
|---|---|---|---|---|---|---|---|
| | | Carnosic Acid | Carnosol | Other Phenolic Diterpenes | Carnosic Acid | Carnosol | Other Phenolic Diterpenes |
| 5 | $2.3 \pm 0.1$ [a] | $72.8 \pm 0.9$ [a] | $100.0 \pm 0.0$ [a] | $100.0 \pm 0.1$ [a] | $75.2 \pm 0.5$ [a] | $70.8 \pm 2.0$ [a] | $88.7 \pm 0.2$ [a] |
| 10 | $3.3 \pm 0.1$ [b] | $44.4 \pm 1.4$ [b] | $49.4 \pm 3.3$ [b] | $30.5 \pm 1.7$ [b] | $73.0 \pm 0.8$ [b] | $60.7 \pm 3.6$ [b] | $65.8 \pm 1.4$ [b] |

The encapsulation of the extracts obtained by acetone or ethanol in water mixtures demands the removal of organic solvents, and possibly concentration of the extract, before mixing the core and wall material and feeding the spray drier. Removal was performed in a vacuum evaporator, as described in Section 2.4, and resulted in partial precipitation of the non-polar materials that were not soluble in water, as well as difficulties in handling the concentrate. This apparently caused some loss of phenolic diterpenes, while carnosic acid might have been further oxidized to carnosol under the high temperature (140 °C) and air contact at the spray drier. Thus, the carnosic acid encapsulation yield was low, while carnosol's exceeded 90% (Table 4).

GA addition improved the results, compared to MD used alone as wall material, especially for carnosic acid and total flavonoids. When comparing ethanol-water with acetone-water extracts, higher carnosic acid yield values were obtained by encapsulation of the former, possibly because a small amount of ethanol always remains during concentration (due to the azeotrope) and is sufficient for the solution of the compound.

With regard to encapsulation efficiency, the values obtained for carnosic acid and other phenolic diterpenes are very low, especially when MD was used alone as wall material. The addition of GA improved significantly the obtained values, although they remained lower than 50%. Carnosol exhibited higher encapsulation efficiency, possibly because some amount of the carnosic acid present in the capsules may have been oxidized to carnosol. These results indicate that the encapsulation of mixtures of polar and non-polar antioxidants was not successful and, therefore, were not further tested during storage.

In order to encapsulate the phenolic diterpenes efficiently, and without high losses during the procedure, they were transferred to an oil carrier, with the simultaneous total acetone removal. MCT was used as the oil carrier because (i) it is stable against oxidation –containing saturated medium-chain fatty acids, (ii) has low viscosity, and (iii) is commonly used in the pharmaceutical and cosmetic industries [29,30]. By thoroughly mixing an appropriate amount of the acetone extract with MCT oil, and evaporating under vacuum, a transparent oil solution was obtained. The ratio of acetone extract to oil was selected so as to obtain a concentration of 2.50 g carnosic acid plus carnosol/kg oil solution. However, during the evaporation of acetone, some losses of phenolic components might have occurred. For this reason, it was considered important to carry out an HPLC-DAD analysis in order to quantify the concentration of phenolic diterpenes in the oil solution. The results showed the following values: 1.82 g carnosic acid/kg oil, 0.55 g carnosol/kg oil, and 0.21 g other phenolic diterpenes/kg oil. The oil solution was used as core material in two different concentrations in the spray drying tests. MD:GA, 2:1, was used as wall material because, together with 4:1, it provided high yield and efficiency for the encapsulation of the water extract, and GA has emulsifying properties; thus, it could help in the formation of a good emulsion. The results, presented in Table 5, show that, when 5% core material was used, the encapsulation yield of carnosic acid and other phenolic diterpenes was increased, and the encapsulation efficiency was almost doubled. However, when core material was raised to 10%, although the encapsulation efficiency remained good, the encapsulation yield was lower than 50% for all components. These results indicate that, with 10% core content, the wall material is not enough to cover all the oil droplets during the emulsification. Thus, a high percentage of the thermosensitive phenolic diterpenes is lost during the vaporization in the spray dryer.

### 3.2.2. Storage Stability of the Encapsulated Products

The encapsulated products were examined for their stability during storage at room temperature. The powders obtained with water extract were stored up to six months, and the loss in phenolic compounds, as well as the changes in encapsulation efficiency, are presented in Figures 1 and 2, respectively. It is evident that the content of phenolic compounds decreases as storage time increases, although differences were minor in one month of storage. After six months, the total flavonoids presented the highest loss, and their content decreased by 25.4–51.0%. Since the encapsulation efficiency did not present significant changes in most cases (Figure 2), it can be assumed that a slow migration of the phenolic components through the wall material takes place, and, being exposed to air contact at the surface of the capsules, they oxidize to several products. The composition of the wall material did not affect either the retention of phenolic compounds or the encapsulation efficiency, except total flavonoids that presented lower retention when MD:GA 2:1 was used. Similarly, the spray drying temperature had no effect.

The powders obtained with acetone extract were stored for one month, and Figure 3a,b present the loss of phenolic compounds and the encapsulation efficiency changes, respectively. It is evident that carnosic acid presented the highest loss during storage. In particular, when core content was equal to 5%, the carnosic acid was reduced by 35.3%, and, when core content raised to 10%, the carnosic acid present in the powder was reduced less and, specifically, by 30.7% in one month of storage. However, in 10% core content, the encapsulation efficiency was reduced to about 28.2%, in contrast to the powder with 5% core content, where the encapsulation efficiency of carnosic acid was reduced only by 3.0%. Carnosol and other phenolic diterpenes did not present significant changes or even seem to augment in some cases, a fact that should be attributed to their formation through the carnosic acid oxidation.

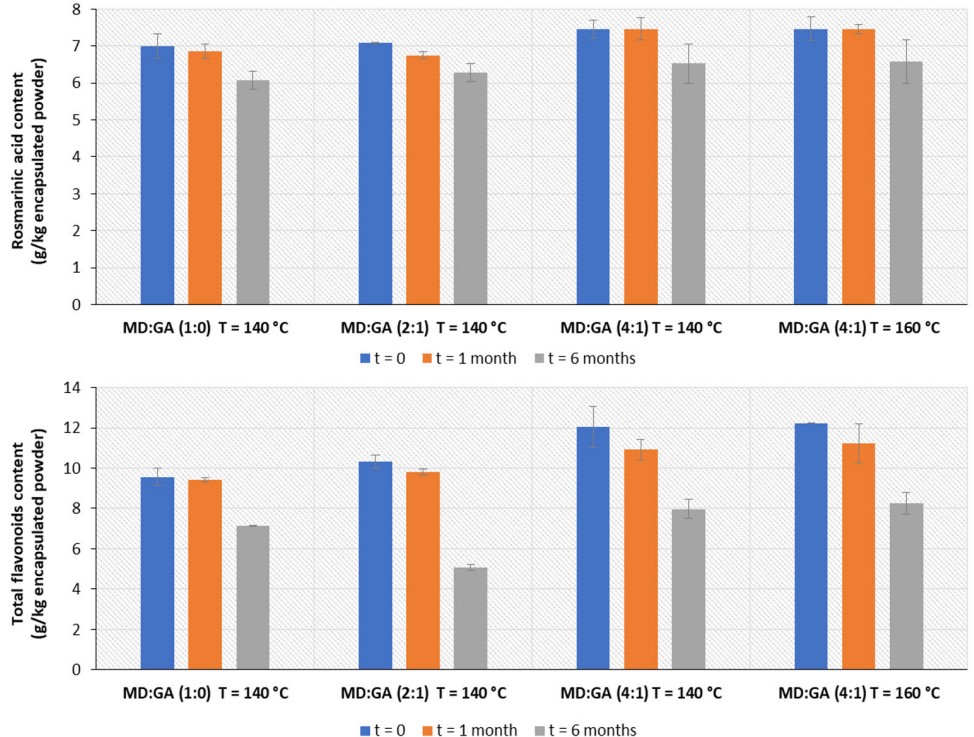

**Figure 1.** Reduction of rosmarinic acid and total flavonoids content over 6 months storage of the encapsulated water extracts. Effect of maltodextrin gum arabic ratio (MD:GA) in the wall material, and inlet air temperature, when core content was maintained constant at 10% and total solids content in the feed mixture 30%.

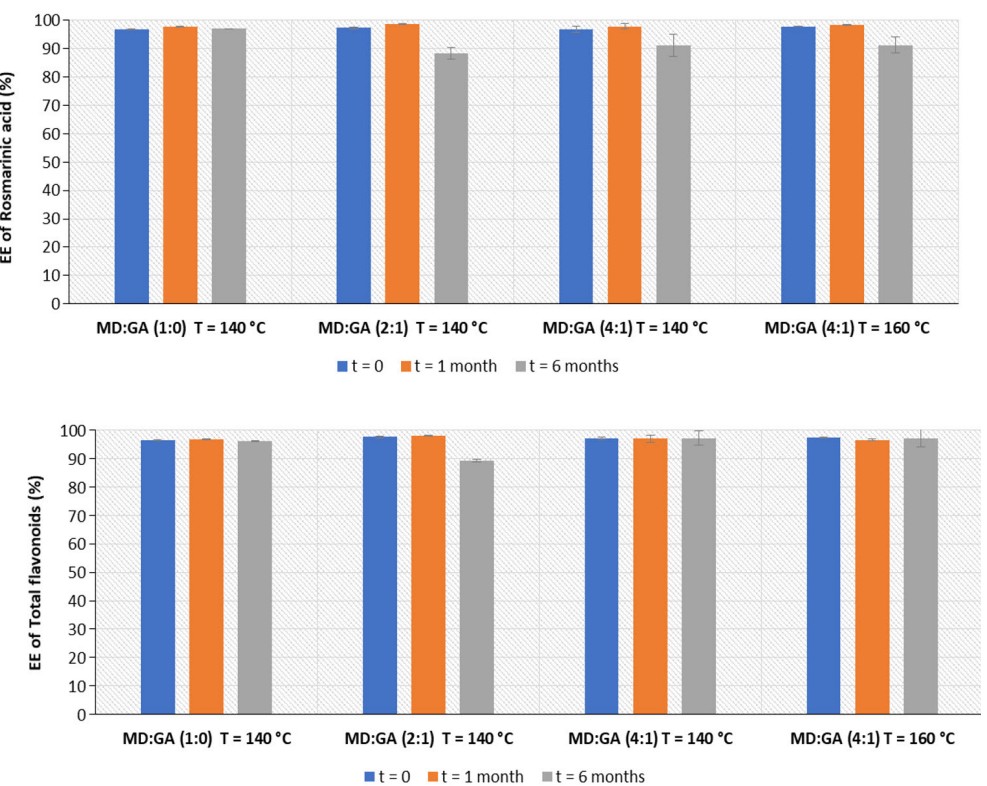

**Figure 2.** Change in encapsulation efficiency of rosmarinic acid and total flavonoids over 6 months storage of encapsulated water extracts. Spray drying parameters as in Figure 1.

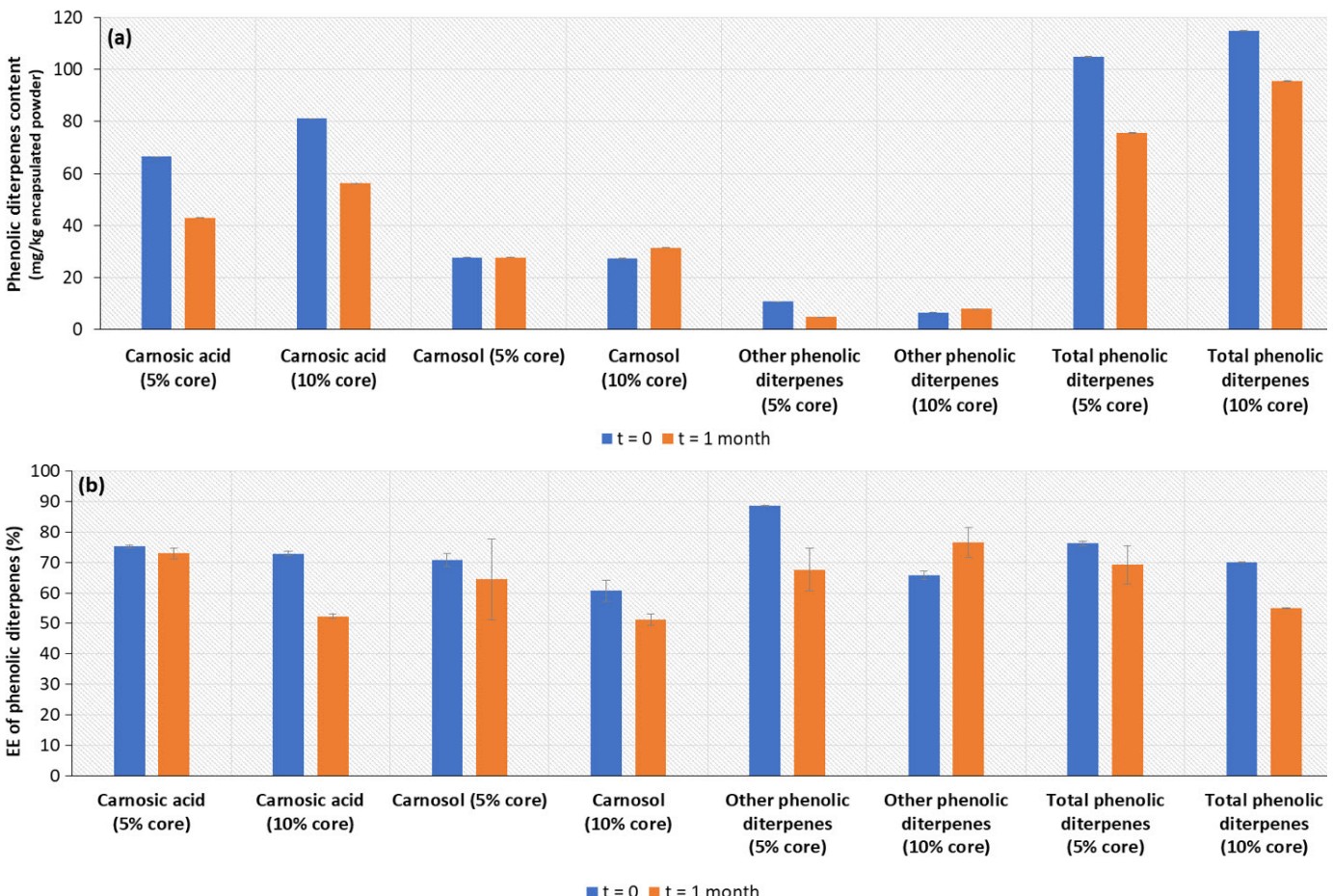

**Figure 3.** (**a**) Reduction of phenolic diterpenes and (**b**) change in their encapsulation efficiency over 1 month of storage of the encapsulated powder containing the acetone extract. Spray drying parameters: core content 5 or 10%, wall material MD:GA 2:1, inlet air temperature 140 °C, and total solids content in the feed 30%.

Overall, the total phenolic diterpenes were reduced by 16.8% when core content was 10%, and 28.0% when core content was equal to 5%.

The encapsulated powder also contains rosmarinic acid and flavonoids that were present in the acetone extract (Table 2). The reduction of these compounds during the one-month storage of the powder was not significant, and the encapsulation efficiency remained constant (data not presented).

### 3.3. Formulation of Emulsion Rich in Phenolic Diterpenes and Storage Stability of the Product

As the yield and efficiency of diterpenes encapsulation was low compared to the rest of phenolic compounds, another formulation procedure was attempted for the acetone extract that was rich in these less polar compounds. Dissolution of the extract in a neutral oil carrier, such as MCT, followed by acetone removal, as described in Section 2.4, resulted in a clear oil solution with the following content: carnosic acid 1.82 g/kg oil carnosol 0.55 g/kg oil and other phenolic diterpenes 0.21 g/kg oil, as mentioned in Section 3.2.1. This oil solution was used to prepare an emulsion with 80 g oil phase/kg emulsion, as described in Section 2.5. The emulsion was stable in structure during storage for 21 days at either 15 or 37 °C. However, a decrease in the phenolic content was observed, as indicated in Figures 4 and 5.

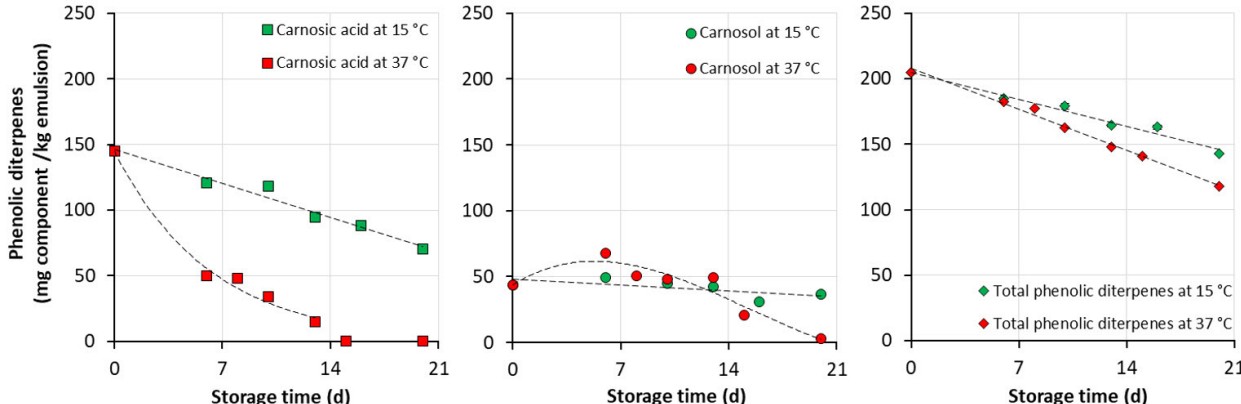

**Figure 4.** Degradation of carnosic acid, carnosol, and total diterpenes in emulsions stored at 15 and 37 °C.

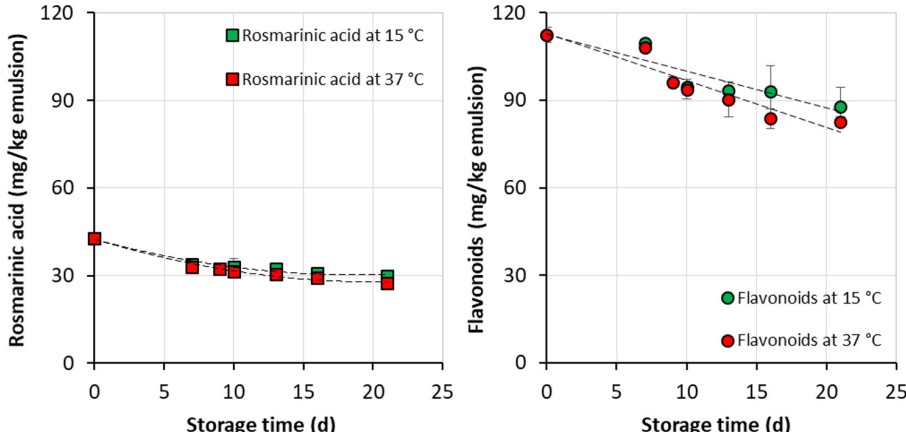

**Figure 5.** Degradation of rosmarinic acid and total flavonoids in emulsions stored at 15 and 37 °C.

As expected, at higher temperature, the decrease rate of carnosic acid was higher. Carnosic acid was firstly converted to carnosol and, later, to other phenolic diterpene derivatives [31]. In just 15 days, carnosic acid had been totally converted to phenolic diterpene derivatives, at the temperature of 37 °C. On the other hand, the same compound, at 15 °C, had been degraded by 52% on the 20th day. Carnosol shows a fluctuation in its concentration, due to the more complex conversions of carnosic acid in it but also its conversion to other secondary phenolic diterpenes, such as methyl carnosate. Thus, for the emulsions that were stored at 37 °C, initially, an increase in carnosol was observed until the 6th day, due to the conversion of carnosic acid in it. The degradation starts then, and, from the 13th day onwards, the degradation was extremely fast and reached the percentage of 93% the 20th day. With regard to the emulsions that were stored at 15 °C, carnosol, for the same reasons, had a fluctuation in its concentration, but the degradation of this compound at the 20th day of the experiment only reached 16%. The concentration of total phenolic diterpenes in both emulsions decreased, but the rate of this degradation was much faster at the high temperature.

Figure 5 presents the degradation of rosmarinic acid and total flavonoids that are also present in the acetone extract (Table 2). The rate of this degradation in both rosmarinic acid and total flavonoids was higher at 37 °C than at 15 °C. The percentage of the degradation of rosmarinic acid at the end of the experiments was 36% and 30% at 37 and at 15 °C, respectively. For the total flavonoids, the degradation reached the percentage of 27% and 22%, respectively.

## 4. Discussion

*4.1. Extraction Yield and Phenolic Compounds Content in the Extracts Obtained by Different Procedures*

Rosemary contains both non-polar, mainly carnosic acid, and polar antioxidants, with rosmarinic acid being the most abundant among them. Both carnosic and rosmarinic acid exhibit very good antioxidant activities and, thereby, several beneficial health effects [32,33]. A quantitative extraction of carnosic acid can be obtained by acetone, while that of rosmarinic acid can be obtained by water or other polar solvents, such as methanol [6]. However, the highest extraction yield, considering the sum of antioxidant compounds, can be achieved by acetone-water, or ethanol-water, mixtures. In particular, acetone 60–80%, *v/v*, and ethanol 50–70%, *v/v*, have shown the highest yields, while a further increase in water content sharply decreases the recovery, especially of carnosic acid [6–8]. The oxidation of carnosic acid to carnosol and other derivatives is favored by the presence of water in the solvent [7,34]. Our results indicated that the highest total phenolic diterpenes yield was obtained by acetone:water 80:20, in accordance with previous papers [7,8]. Additionally, the application of UAE increased the recovery of antioxidant compounds or decreased the extraction time, due to cavitation phenomena that enhanced the penetration on the solvent into the plant matrix [9]. The results of our experiments, employing UAE with acetone:water 80:20, *v/v*, and ethanol:water 60:40, *v/v*, are in general agreement with other published results [7,8]. However, the use of ethanol:water 60:40 in UAE resulted in partial oxidation of carnosic acid to carnosol and decrease of the overall phenolic diterpenes yield.

Moreover, we attempted a different extraction procedure, comprising a fixed bed of the plant material, through which the solvent was continuously passed. This semi-batch extraction in a fixed bed reactor has several advantages. It is carried out in environmentally-friendly conditions (room temperature) and does not require separation of the extract from the solid, thus reducing the cost of extraction, and facilitating its scale up for industrial application. In addition, there is a difference in the solute concentration between the solid matrix and the liquid phase throughout the extraction. The mass transfer potential from a solid to a liquid phase is the concentration difference. As long as a pure solvent is continuously circulating in the extractor, the mass transfer potential will never be zero, and the extraction will take place until the solid is depleted, as opposed to the conventional solid/liquid extraction, where the solvent is not renewed, and equilibrium is reached. The extraction may be continued until the solute in the plant matrix is exhausted.

By passing 200 mL of acetone:water 80:20, through a bed of 20 g plant material, we obtained similar recovery of phenolic compounds with UAE, performed with double the liquid to solid ratio. It should be also noted that the obtained extract had a higher antioxidant activity than that obtained by UAE. Sonication generates hydroxyl radicals that probably oxidize some of the phenolic compounds present in the extract. Another advantage of the fixed bed extraction is the ability of performing successive extractions with different solvents, without the need of separation the solid material from the liquid and reloading it to the bed. Thus, the active components of the plant can be separated according to their solubility in the different solvents. Rosemary is known for its high content in carnosic acid and carnosol, which, given their potent antioxidant activity, are used to valorize the commercial rosemary extracts. Based on this, we performed a first extraction with acetone, to recover carnosic acid and its derivatives, followed by an extraction with water to recover the most polar phenolic compounds. An acetone extract, rich in carnosic acid (9.5 g/kg dry extract) and derivatives (3.6 g/kg dry extract), was obtained, though the yield on dry plant basis was lower than that obtained by the acetone-water mixture. This should be attributer either to the liquid volume that passed through the bed that was not enough to exhaust all the compounds, or to the ability of water to wet and open the plant structure, thereby enabling a better penetration of the solvent and more quantitative extraction of the compounds.

The extraction selectivity obtained by mixtures of organic solvents with water was higher than that obtained by water, and even more by acetone. This means that pure

acetone may have extracted several non-phenolic compounds, such as waxes. The removal of these compounds enables the better penetration of water, in the following extraction, and its contact and dissolving of polar compounds, e.g., sugars, in addition to polar phenols. On the other hand, either pure acetone or pure water cannot extract as many phenolic compounds as the mixtures of solvents with water (Table 2). Thus, the selectivity obtained by the former is lower. Literature data confirm that the selectivity of pure acetone or water is lower than the obtained with mixtures of organic solvents (acetone or ethanol) with water [7,8].

Nevertheless, both extraction techniques (FBE or UAE) can be easily scaled up, for industrial applications. FBE can be used to effectively separate non-polar antioxidants that can be incorporated in lipid substrates, from more polar ones that may be applied to high moisture food or cosmetics. However, many formulations contain both lipid and aqueous phase, and, thereby, a rosemary extract rich in both carnosic and rosmarinic acid (the most potent antioxidants of the herb), as well as other phenolic diterpenes and flavonoids, might be the most protective for such formulations. In that sense, either FBE with mixtures of acetone or ethanol with water could be applied, or UAE, which presents similar results in terms of antioxidants recovery but has the benefit of shorter processing time, that is important for industrial application.

The tentative identification of the compounds present in all extracts was based on retention time and spectral data, as described in detail by Psarrou et al. [8]. In addition to the flavonoids detected and identified in that study, one more compound was eluted after rosmarinic acid and before isoscoutelarein (4′,5,7,8-tetrahydroxyflavone), during the chromatographic analysis of our extracts. The UV spectrum of the compound showed $\lambda_{max}$ at 272 and 334 nm. Cuvelier et al. [35] obtained a similar chromatogram to our study and identified the compound eluting between rosmarinic acid and isoscoutelarein as hispidulin 7-O-glucoside or homoplantaginin. According to literature data, homoplantaginin was identified with main peaks at 273 and 334 nm [36,37]. We, therefore, assume that the peak eluted after rosmarinic acid most probably concerns the homoplantaginin.

### 4.2. Encapsulation of the Extracts by Spray Drying and Storage Stability of the Products

Among the numerous potent wall materials that can be used for encapsulation, MD is the most frequently applied in spray drying because it has low cost, high solubility in water, ability to form films, and low viscosity even at high concentrations in solutions [19,38]. Blending of MD with other coating materials improves the encapsulation efficiency due to the different structure of the materials that causes denser bonding. In particular, GA is the most commonly used carrier of lipophilic compounds due to its excellent emulsifying ability, which is attributed to its hydrophobic protein fraction that is covalently linked to hydrophilic polysaccharide structures [39]. Our experimental results indicated that the combination of MD and GA at a ratio 4:1 or 2:1 increased the encapsulation yield and efficiency of rosmarinic acid, and even more of the less polar flavonoids, although with no significant differences in some cases. Garofulic et al. [40] observed that different phenolic compounds showed different retention in the encapsulated powder, and phenolic acids presented higher values than anthocyanins and flavonoids. In addition, spray drying of a double (water-oil-water) emulsion of rosemary, with maltodextrin as wall material, indicated that the phenolic acids (rosmarinic and caffeic) present better retention than the flavonoids, especially luteolin, which was the most heat sensitive, according to the authors [41]. Moreover, each class of compounds presented different results when entrapped in different wall materials, with phenolic acids showing better results in MD encapsulation and flavonoids in GA [40]. These observations indicate that, in addition to thermal sensitivity that may cause higher losses of the thermal sensitive compounds, the interactions with the wall material play an important role. GA is a heteropolysaccharide containing approximately 2–5% protein, which has a hydrophobic character, opposite to carbohydrates that are hydrophilic [15,42]. Thus, the incorporation of GA in MD facilitates the interaction with the hydrophobic flavonoids of the extracts and, consequently, their encapsulation yield

and efficiency. The obtained encapsulation yield values with MD and GA combinations were, in all cases, higher than 90% (Table 3). Tolun et al. [18] also observed that the ratio MD:GA of 4:1 gave the highest content of grape phenolics in the encapsulated powder, compared to 1:0 or 3:2 ratios.

Additionally, the encapsulation efficiency varied around 97% (Table 3), indicating a very good protection of the antioxidant compounds. These values are similar to those reported in literature for grape phenolics [18], and higher than those reported for anthocyanins [28,43].

In general, the encapsulation yield and efficiency are increased when the core content is decreased because the wall material concentration is high and enables better engulfing of the core. A core content of 10% gave the best results in the encapsulation of phenolic compounds [19,28] and was, therefore, selected in our experiments.

The increase of inlet air temperature may cause degradation of the phenolic components at the spray drier, possibly cracking of the capsule surface, thus decreasing the encapsulation yield [18,19]. However, the increase from 140 to 160 °C did not cause a decrease when the core content was as low as 10% [19], in agreement with our results. This should be attributed to the high protection of the core material by the wall material that has a high concentration and, thus, forms a thick layer.

The moisture content of the obtained powder increased by the addition of GA (Table 3) as it has more hydrophilic groups than MD and, thereby, may absorb more water [44]. Different wall materials present different hydroscopicity, and, in that sense, the use of MD with a high DE or higher amounts of GA would increase the moisture content of the final product. In addition, the moisture content depends on the wall material concentration and is generally higher, as this concentration increases due to higher hydroscopicity of the wall material compared to core. Another process variable that affects moisture content is the inlet air temperature and, consequently, outlet temperature of the spray drier. Higher inlet air temperature decreases the moisture content [45]. Low moisture content is desirable not only to assure microbial stability but also to avoid agglomeration of the product [19]. However, in all cases, the moisture content of the obtained powders did not exceed the value of 4.2%, thus assuring the microbial stability of the product [44,46].

The extracts obtained by mixtures of organic solvents with water showed encapsulation yield and efficiency similar to water extracts for rosmarinic acid and total flavonoids but much lower yield for carnosic acid and other phenolic diterpenes, as well as lower efficiency for all the phenolic diterpenes (Table 4). The lower yield is probably due to thermal sensitivity of carnosic acid, which is oxidized to carnosol during the solvent evaporation, the emulsification, and the spray drying. The lower efficiency may be attributed to the non-polar structure of the phenolic diterpenes, which does not promote the formation of intermolecular forces and interactions with the wall material and, thereby, the entrapment in the wall material. This is justified by the fact that the lowest efficiency of carnosic acid encapsulation was observed when MD alone was used as wall material, which shows a hydrophilic character.

In order to protect the phenolic diterpenes and increase their encapsulation yield and efficiency, they were transferred to an oil solution, as described in the experimental section. This procedure has not been reported in literature, while the encapsulation of carnosic acid and carnosol to obtain a powder has not been attempted by any researcher, according to our knowledge. A higher amount of GA was used in the wall material (MD:GA equal to 2:1), so as to increase the hydrophobic character and the emulsifying properties. Thus, a better emulsification and entrapment of the compounds was achieved, with encapsulation yield and efficiency of the carnosic acid amounting to 72.8% and 75.2%, respectively, when the core content was 5% of total feed solids (Table 5). Higher core content (10%) was not effectively encapsulated, as the wall material was not adequate to cover the oil droplets and entrap them into the capsules. Nevertheless, with a core content of 5%, a powder containing 94 mg carnosic acid plus carnosol/kg powder, and 105 mg total phenolic diterpenes/kg powder was obtained.

The storage stability study indicated minor losses of rosmarinic acid and total flavonoids, during one month of storage, while flavonoids presented the highest reduction after six months. This may be attributed either to the better protection of rosmarinic acid, as it is more hydroscopic and, thereby, strongly bonded by the wall material, or to the higher degradation rate of flavonoids, compared to rosmarinic acid. Mahdavi et al. [43], studying the degradation of anthocyanins encapsulated in MD or MD:GA, during 90 days of storage, found a more rapid reduction than the observed in our work. Anthocyanins are very sensitive components and degrade rapidly during storage. Additionally, Moser et al. [47] observed different retention of anthocyanins, flavonols, hydroxybenzoic acid derivatives, and flavan-3-ols during 150 days of storage, which also depended on the wall material composition (soy or whey protein combined with MD) and core-to-wall ratio. These results indicate that the storage stability is related to the degradation rate of the encapsulated compounds. In our experiments, the composition of the wall material did not affect the phenolic compounds decrease, as also reported by Mahdavi et al. [43] for the reduction rate of anthocyanins, a fact that should be attributed to the equal encapsulation efficiency that was obtained by using different MD:GA ratios. However, it should be noted that a higher decrease of flavonoids was observed in the encapsulated powder with MD:GA 2:1, which needs further investigation.

The powders with the encapsulated acetone extract were stored for a shorter time (one month) as their main ingredients, i.e., phenolic diterpenes, are easily degraded. In fact, carnosic acid decreased by 30–35% in one month of storage (Figure 3a). The encapsulation efficiency of carnosic acid was initially similar (75% and 73%) when either 5% or 10% of core material was used for the encapsulation. As presented in Figure 3b, after one month of storage, the powder obtained with 5% core material maintained its encapsulation efficiency, while it was reduced to around 28% in the powder obtained with 10% core. This supports the assumption that, when 10% core material was used, the quantity of the wall material was not adequate and formed a thinner coating of the oil droplets. Thus, carnosic acid can diffuse easier through the wall material to the surface of the capsules during storage, and the encapsulation efficiency is reduced. The changes observed in carnosol and other phenolic diterpenes, after one month of storage, do not lead to accurate conclusions because these compounds are formed through the oxidation of carnosic acid and also degrade to other oxidation products. Taking into account that not only carnosic acid but also its derivatives possess antioxidant activity [48,49], we can assume that the obtained powder has an appreciable antioxidant activity after one month of storage.

The powders containing the extracts obtained by mixtures of organic solvents with water were not subjected to storage stability experiments because the results of the encapsulation efficiency of phenolic diterpenes were not satisfactory. The low encapsulation efficiency at zero time implies that the majority of the phenolic diterpenes are located on the outside part of the capsule and will degrade very soon, due to their contact with the environment. For this reason, different handling of extracts from solvent mixtures is proposed and, in particular, their incorporation into an oily carrier in order to achieve better encapsulation.

### 4.3. Storage Stability of Emulsions Rich in Phenolic Diterpenes

Although several studies have examined the antioxidant protection offered by plant extracts to oil-in-water emulsions, only a few have investigated the stability of the natural phenolic antioxidants that are present in the extracts. Natural phenolic compounds have several health benefits, but they are limited due to their low stability and bioavailability in the digestion and absorption process [50]. Additionally, most of these compounds are not dissolved in water and, therefore, cannot be easily incorporated in water-based foods or cosmetics. The delivery of these compounds in an emulsified form can solve some of the aforementioned problems. In particular, MCT, as the oil carrier of bioactive compounds in oil-in-water emulsions, extended the final digestion of the lipid phase and

the bioaccessibility of the bioactive compound, compared to long or short chain fatty acids triacylglycarols [51].

The carnosic acid concentration in the emulsion decreased faster than that of all other phenolic compounds. It is well-known that carnosic acid is a very effective antioxidant, easily oxidized to carnosol and other derivatives [48,49]. As a compound of medium polarity, carnosic acid is rather distributed in the oil-water interface, while the less polar carnosol, methyl carnosate, and other oxidative derivatives are preferably located in the lipid core of the emulsion droplets [49], which is better protected from oxidation. Several researchers [52,53] examined the antioxidant activity in relation to the partition of the phenolic compound in the aqueous phase, oil phase, or interface. The authors concluded that the more phenolic antioxidant is distributed in the interfacial film, the stronger the antioxidant activity will be in the emulsion system. Therefore, carnosic acid may be easily oxidized, as it is present at the interface.

Carnosic acid in a Tween-based emulsion disappeared after one-day storage at accelerated conditions (60 °C), while 2% of the initial methyl carnosate content remained after two days [49]. Higher stability of carnosic acid was observed when it was encapsulated in Tween- and lecithin-based nanoemulsions [54], while its bioaccessibility and bioavailability were remarkably improved in lecithin-based nanoemulsions [55].

The rest of antioxidant compounds present in the rosemary extract showed lower degradation rates than carnosic acid (Figure 5). Rosmarinic acid decreased faster than the total phenolics, in agreement with the results of Choulitoudi et al. [56]. Rosmarinic acid is a polar compound and is, thereby, distributed mainly in the aqueous phase and the oil-water interface, where it can be oxidized more promptly by trace metals or free radicals. The flavonoids, which are present in the acetone extract (Table 2), are mostly non-polar and stable against oxidation, since they do not possess many hydroxyl groups in positions favoring antiradical activity. Among the eight flavonoids of the acetone extract nepitrin, isoscutellarein and ladanein possess active antiradical hydroxyls, i.e., the *o*- or/and *p*-hydroxyls. These compounds represent the considerable 43% of the total flavonoids in the acetone extract; however, only nepetrin is glycosylated and, thus, more polar and more likely to be located at the interface.

## 5. Conclusions

According to the results presented and discussed by the authors in the previous sections, it can be concluded that:

- the water rosemary extracts can be very effectively encapsulated in maltodextrin combined with gum arabic with high encapsulation yield (90–100%) and efficiency (97%) for rosmarinic acid and flavonoids;
- the acetone extracts—rich in carnosic acid and carnosol—should be first transferred to an oil solution and then encapsulated as dry powder or emulsion;
- over a period of six months of storage of the encapsulated products, a high retention of rosmarinic acid (88%) and lower of flavonoids (50–75%) were observed; and
- carnosic acid presented lower retention either encapsulated in solid powder (65–70% after one month at ambient temperature) or in emulsion (48% after 20 days of storage at 15 °C), along with its partial conversion to carnosol.

The findings of this study could contribute to ongoing research in the scientific field of encapsulation of natural phenolic antioxidants, thereby leading to market ideas of interest for food, cosmetic, and pharmaceutical industrial sectors.

**Author Contributions:** Conceptualization, V.O. and D.T.; methodology, V.O. and D.T.; software, L.-D.K.; validation, L.-D.K., D.T., and V.O.; investigation, L.-D.K.; resources, L.-D.K., D.T., and V.O.; data curation, L.-D.K. and D.T.; writing—original draft preparation, L.-D.K.; writing—review and editing, V.O. and S.K.; supervision, V.O.; project administration, V.O. The views expressed in this publication are purely those of the writers and may not in any circumstances be regarded as stating

an official position of the European Commission. All authors have read and agreed to the published version of the manuscript.

**Funding:** This research received no external funding.

**Institutional Review Board Statement:** Not applicable.

**Informed Consent Statement:** Not applicable.

**Conflicts of Interest:** The authors declare no conflict of interest.

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
