# Peer review of "Formulation of Rosemary Extracts through Spray-Drying Encapsulation or Emulsification"

_nutraceuticals, doi:10.3390/nutraceuticals2010001_

Round 1

Reviewer 1 Report

The manuscript entitled “Formulation of rosemary extracts through spray-drying encapsulation or emulsification” outlines a thorough approach to the potential use of rosemary antioxidants, including their extraction from plant, purification, and formulation processes to convert them to a stable solid state. Study is well performed and organized, obtained results are promising to be implemented in real settings of production. Overall, I suggest accepting the proposed paper.

English language and style are fine/minor spell check required. Here are some constructive suggestions:

  • The introduction is too general. It should be rewritten to point out the importance of rosmarinic acid use, as well as the advances of use of spray drying in combination with adequate carriers, i.e maltodextrin and gum Arabic (or just replace lines 571-577 from discussion to intro in some extent)
  • Statements in lines 43-51 must be followed by the relevant citations
  • Line 122 : what was the solvent flow?
  • Lines 146-156 should be more concise in means which order of the steps was. Was the solvent firstly removed from the extracts or not?
  • Are the process parameters during spray drying, such inlet air temperature chosen according to the preliminary experiments or literature data. Please elaborate
  • HPLC-DAD analysis should be explained more in details
  • What was the size distribution of the obtained encapsulates?
  • Lines 299-303: regarding the explanation on the selectivity of solvents used, has been it already demonstrated elsewhere? These sentences should be moved to discussion section
  • Results section should be reserved for clear data description only. Thus, comparison with the literature data in lines 315-319 should be moved to Discussion section.
  • In section 4.1. it would be nice that authors give some comment with the regards of scaling -up the extraction techniques used
  • Line 554, decimal separator should be point not comma
  • Line 599 correct the typo in word phenolics

Author Response

The manuscript entitled “Formulation of rosemary extracts through spray-drying encapsulation or emulsification” outlines a thorough approach to the potential use of rosemary antioxidants, including their extraction from plant, purification, and formulation processes to convert them to a stable solid state. Study is well performed and organized, obtained results are promising to be implemented in real settings of production. Overall, I suggest accepting the proposed paper.

English language and style are fine/minor spell check required. Here are some constructive suggestions:

The manuscript has been checked and some corrections were made

  • The introduction is too general. It should be rewritten to point out the importance of rosmarinic acid use, as well as the advances of use of spray drying in combination with adequate carriers, i.e maltodextrin and gum Arabic (or just replace lines 571-577 from discussion to intro in some extent)

We would like to thank the reviewer for this suggestion. The introduction was improved, emphasizing on the importance of rosmarinic acid and other antioxidant components of rosemary and on the advantages of spray drying, as suggested.

  • Statements in lines 43-51 must be followed by the relevant citations

Citations have been added

  • Line 122 : what was the solvent flow?

2 mL/min, added in the revised text.

  • Lines 146-156 should be more concise in means which order of the steps was. Was the solvent firstly removed from the extracts or not?

The text has been revised so as to clarify the removal of the water or solvents and adjustment to final volume.

  • Are the process parameters during spray drying, such inlet air temperature chosen according to the preliminary experiments or literature data. Please elaborate

The process parameters during spray drying were selected according to literature and preliminary experiments. Details have been added in the revised text.

  • HPLC-DAD analysis should be explained more in details

Details were added in the revised text

  • What was the size distribution of the obtained encapsulates?

We did not measure the size of the obtained encapsulates. We agree with the reviewer that the size and size distribution is an important quality attribute of the encapsulated product but we did not have the appropriate infrastructure for its measurement. We hope to proceed with such measurements in future experiments.

  • Lines 299-303: regarding the explanation on the selectivity of solvents used, has been it already demonstrated elsewhere? These sentences should be moved to discussion section

Selectivity has been shortly discussed in literature [7,8]. The comments about selectivity were moved to the discussion section as suggested, and comparison with literature reports was added.

  • Results section should be reserved for clear data description only. Thus, comparison with the literature data in lines 315-319 should be moved to Discussion section.

All comparison with the literature data was moved to the discussion, as suggested.

  • In section 4.1. it would be nice that authors give some comment with the regards of scaling -up the extraction techniques used

We would like to thank the reviewer for this suggestion. Some comments about the scale up and benefits of the extraction techniques were added in the revised text.

  • Line 554, decimal separator should be point not comma

Corrected

  • Line 599 correct the typo in word phenolics

Corrected

Reviewer 2 Report

The study aims to define the optimal solid/liquid extraction and formulation of rosemary extracts conditions for antioxidants recovery, in terms of total phenolic content as well as individual phenolic compounds. The subject of the work is interesting, especially that in addition to the total content of polyphenols, the content of individual compounds was also assessed using the chromatographic method. The obtained results are also adequately described and discussed with previously published works. Detailed comments are provided below.

In the introduction of the work, attention should be paid to the innovative nature of the research presented in the article. According to the presented earlier publications, there are many reports specifying the formulation of plant extracts through spray-drying encapsulation or emulsification. What is the innovative aspect of the research described? What information does this research bring to the field of science about the formulation processes? This aspect should be discussed in the introduction to the research. I also recommend adding at the end of the introduction the purpose of the research to be carried out in the manuscript, but not specifying in the introduction what were the results of the research.

The methods of producing individual products (extracts, powders, emulsions) are described in detail and it is possible to repeat the experiment. The design of the experiment is also described, in how many repetitions the experiment and analyzes have been repeated. I have no comments to this part, only I recommend using SI units in the description of the emulsion preparation method. I also recommend adding the conditions used in the chromatographic determination. The description does not include the analysis temperature, eluents and the description of the gradients, if the determination was performed using the gradient method.

The description of the standards used in HPLC analysis is given carnosic acid, carnosol, rosmarinic acid and quercetin dihydrate. Unidentified compounds belonging to the flavone or flavonol subgroups are reported to be quantified as quercetin equivalents.  In contrast, the results presented in Table 2 relate to the quantification of other compounds. How was the content of flavonoids and caffeic acid calculated?

Why are there significant differences not specified in the column of Powder moisture EE (%) in Table 4?

Author Response

The study aims to define the optimal solid/liquid extraction and formulation of rosemary extracts conditions for antioxidants recovery, in terms of total phenolic content as well as individual phenolic compounds. The subject of the work is interesting, especially that in addition to the total content of polyphenols, the content of individual compounds was also assessed using the chromatographic method. The obtained results are also adequately described and discussed with previously published works. Detailed comments are provided below.

  • In the introduction of the work, attention should be paid to the innovative nature of the research presented in the article. According to the presented earlier publications, there are many reports specifying the formulation of plant extracts through spray-drying encapsulation or emulsification. What is the innovative aspect of the research described? What information does this research bring to the field of science about the formulation processes? This aspect should be discussed in the introduction to the research. I also recommend adding at the end of the introduction the purpose of the research to be carried out in the manuscript, but not specifying in the introduction what were the results of the research.

We would like to thank the reviewer for this suggestion. The introduction has been extended and the innovation as well as the purpose of the research were added in the last paragraph. The main results have been briefly described in the original manuscript, according to the guidelines of the Journal but were removed from the revised manuscript as suggested by the reviewer.

  • The methods of producing individual products (extracts, powders, emulsions) are described in detail and it is possible to repeat the experiment. The design of the experiment is also described, in how many repetitions the experiment and analyzes have been repeated. I have no comments to this part, only I recommend using SI units in the description of the emulsion preparation method. I also recommend adding the conditions used in the chromatographic determination. The description does not include the analysis temperature, eluents and the description of the gradients, if the determination was performed using the gradient method.

SI units were used in the emulsion preparation, and details about the HPLC analysis were added in the revised text.

  • The description of the standards used in HPLC analysis is given carnosic acid, carnosol, rosmarinic acid and quercetin dihydrate. Unidentified compounds belonging to the flavone or flavonol subgroups are reported to be quantified as quercetin equivalents.  In contrast, the results presented in Table 2 relate to the quantification of other compounds. How was the content of flavonoids and caffeic acid calculated?

We would like to thank the reviewer for this remark. Caffeic acid was used as a standard (added to standards in the revised text), while all flavonoids were quantified as quercetin equivalents as indicated in the revised text.

  • Why are there significant differences not specified in the column of Powder moisture EE (%) in Table 4?

The values are the same as for EY (%), where significant differences were indicated. To avoid any confusion we added letters indicating significant differences in the revised text.